# Interactive Large Language Models for Reliable Answering under Incomplete Context

**Jing-Cheng Pang**[1,3,*], **Heng-Bo Fan**[2,*], **Pengyuan Wang**[1,3,*], **Jia-Hao Xiao**[2,*] **Nan Tang**[1,3],
**Sihang Yang**[1,3], **Chengxing Jia**[1,3], **Ming-Kun Xie**[2], **Xiang Chen**[2], **Sheng-Jun Huang**[2], **Yang Yu**[1,3]
[1] **National Key Laboratory for Novel Software Technology, Nanjing University, China**
**& School of Artificial Intelligence, Nanjing University, China**
[2] **College of Computer Science and Technology/Artificial Intelligence,**
**Nanjing University of Aeronautics and Astronautics, China**
[3]**Polixir.ai**

**Reviewed on OpenReview:** `https://openreview.net/forum?id=nnlmcxYWlV`

## Abstract

The rise of large language models (LLMs) has revolutionized the way humans interact with artificial intelligence systems. However, their reliability in sensitive applications—such as personal consultations or clinical decision-making—remains limited. A critical shortfall lies in LLMs' inherent lack of interactivity: these models generate responses even when essential context or domain-specific knowledge is absent, risking inaccurate or misleading outputs. A potential approach to mitigate this issue is to enable LLMs to pose clarifying questions, thereby uncovering the missing information required to provide accurate responses. However, previous methods often tend to greedily prompt LLMs to ask questions. This burdens the user to respond to potentially irrelevant questions and makes the system less flexible. In this paper, we introduce LaMSeI (Language Model with Selective Interaction) method, which enhances LLMs' ability to judge when interaction is necessary under ambiguous or incomplete contexts. The motivation of LaMSeI is to measure the level of LLMs' uncertainty about the user query, and interacts with user only when the uncertainty is high. Additionally, we incorporate active learning techniques to select the most informative questions from question candidates, for effectively uncovering the missing context. Our empirical studies, across various challenging question answering benchmarks, where LLMs are posed queries with incomplete context, demonstrate the effectiveness of LaMSeI. The method improves answer accuracy from 31.9% to 50.9%, outperforming other leading question-answering frameworks. Moreover, in experiments involving human participants, LaMSeI consistently generates answers superior to or comparable to baselines in more than 82% of the cases. Moreover, we verify the performance of LaMSeI on various LLMs, such as LLAMA2, LLAMA3, Vicuna and GPT-3.5, highlighting its capability to improve interactive language models.

## 1 Introduction

Recent advances in large language models (LLMs) (OpenAI, 2023; Team, 2024a; Du et al., 2022; DeepSeek-AI, 2025; Google Deepmind, 2025) lift the curtain of a new era in human-machine interaction. These language models, pre-trained on massive corpora of text (Gao et al., 2021), are designed to comprehend and respond to user queries with human-like proficiency. One of the most impressive applications of LLMs is the chatbot, which generates responses to various user queries in natural language. However, effective dialog requires sufficient contextual information, which is often omitted when user interact with LLM chatbots. For instance, patients may inadvertently neglect to mention certain symptoms when consulting LLMs about their health,

---

*Equal contribution. Corresponding: huangsj@nuaa.edu.cn.

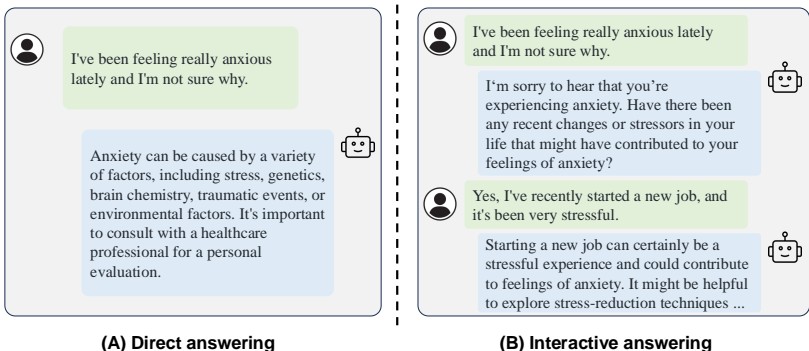

Figure 1: An illustration of interactive LLMs, which seeks for more dialog context to provide helpful answers. (A) LLMs directly generate answer to the user query. (B) LLMs interact with the user to seek more context.

as depicted in Fig. 1 (A). On the other hand, LLMs are trained to respond to various questions and often attempt to provide answers even when context is lacking or their own knowledge is insufficient, which can frequently lead to low-quality responses.

To mitigate the issue of incomplete context information, two principal strategies have been investigated: *Retrieval-Augmented Generation* (RAG (Lewis et al., 2020)) and *asking clarifying questions* (Zhao & Dou, 2024; Andukuri et al., 2024; Tix & Binsted, 2024). RAG introduces the concept of augmenting LLMs with external information sources, such as knowledge bases or search engines (Press et al., 2023). However, these RAG-based methods primarily focus on the the limited knowledge of LLMs and do not adequately address the absence of user context (Wang et al., 2023a). Except for RAG, another solution is *multi-turn clarification*, which asks users clarifying questions to obtain additional context. For example, MulClari-LLM method (Zhao & Dou, 2024) uses a rule-based system to generate a series of clarifying questions in a multiple-choice format, with the aim of clarifying user intent. STaR-GATE (Andukuri et al., 2024) fine-tunes the LLM to ask clarifying questions that are likely to elicit high-quality responses. While these methods effectively enhance the answer quality, they typically operate under a manner that encourage LLMs to ask more clarifying questions, which are often unrelated, resulting in an increasing burden on users to answer them. This calls for a method that enables LLMs to ask clarifying questions in a more flexible and adaptive manner.

In this paper, we address the challenge of bridging the information gap between LLMs and users by introducing the LaMSeI (Language Model with Selective Interaction) method. Fig. 1 (B) illustrates the concept of interactive LLMs. We consider two pivotal questions for enhancing LLM-user interaction: when should LLMs interact with the users, and what questions should LLMs ask to acquire additional context? Firstly, we suggest that LLMs should interact with the LLMs only when they are highly uncertain about the user query. To achieve this, LaMSeI estimates the LLM's uncertainty regarding the query by sampling multiple responses and calculating their variation. Under high uncertainty, LaMSeI prompts the model to generate question candidates regarding the user query, enabling LLMs to pose effective clarifying questions. These candidate questions are not equally effective in bridging the information gap. To deal with this problem, we adopt the active learning technique to select the most informative clarifying questions, aiming to narrow the gap between LLMs and users. As a result, by answering these questions, the user enables the LLM to acquire the missing contextual information.

The main contributions are as follows: Firstly, we underline the critical need for interactive LLMs that flexibly interact with the users to uncover the missing context from the users. We suggest that the clarifying questions should be posed with caution to reduce the users' workload in answering them, *which has been neglected by previous works.* Secondly, we propose a selective interaction mechanism for LLMs using active learning techniques, which aims to select the most informative questions to interact with the user. To our knowledge, this is the first work to combine active learning techniques and LLMs to handle user queries with incomplete context. This technique enables LLMs to dynamically obtain query-specific information. Lastly, we conduct comprehensive experiments to evaluate the algorithms' performance on user queries with imcomplete context, demonstrating that LaMSeI consistently outperforms existing Q & A frameworks on

various LLMs. Furthermore, we evaluate LaMSeI with a wide range of backbone LLMs, demonstrating its efficacy in improving LLM's comprehension of user query, thus enhancing the quality of model response.

## 2 Related Work

**Handling user input with incomplete information.** Previous studies have explored constructing AI systems that better understand user intention, under incomplete information in the user query (Griot et al., 2025; Yang, 2025). A promising approach is to interact with users by posing questions for clarification or information (Aliannejadi et al., 2019; Shi et al., 2022; Zamani et al., 2020; Stoyanchev et al., 2014). Such systems have been effectively implemented in domains like hotel reservation services (Bemile et al., 2014), where they prompt users with specific inquiries to verify and complete booking details. The field of natural language processing has also shown an interest in generating clarifying questions, particularly in response to ambiguous queries. For example, Zamani et al. (2020) utilizes a set of question templates to address the issue of ambiguous web search queries. Stoyanchev et al. (2014) studies the task of selecting clarifying questions from a set of human-generated questions for open-domain information seeking. A separate investigation (Coden et al., 2015) focuses on asking clarifying questions for entity disambiguation, phrased as "Did you mean A or B?" However, this method is limited to entity disambiguation and does not apply to a broader range of queries, such as those with multiple facets. MEDIQ (Li et al., 2024) proposes a dynamic medical consultation framework to simulate the medical consultation process, where doctor repeatedly asks the patient questions for more information. Though various research studies seek user clarification, they generate clarifying questions by rule or prompt LLMs to pose excessive questions. This paper proposes that we should conduct selective interaction and only interact when necessary, improving the overall system efficacy.

**Enhancing LLM answer.** Research on improving LLM response has attracted considerable attention (Zamani et al., 2020; Hu et al., 2025). Related methods can be divided into two categories: invasive methods that train LLM to generate high-quality response (Pang et al., 2024), and non-invasive methods that improve generation by simply providing in-context demonstrations of the task (Brown et al., 2020), or increasing the model's reasoning ability through various answer refinement techniques (Wei et al., 2022; Wang et al., 2023b). However, a considerable limitation in these prompt-based methods is that they cannot elicit knowledge absent from training data, which leads to hallucinated response (Huang et al., 2023), especially in specialized domains (OpenAI, 2023). A related line of approaches improves model response by allowing the model to access external knowledge or information. These proposed methods, known as Retrieval-Augmented Generation (RAG) (Izacard et al., 2023; Wu et al., 2025; Mialon et al., 2023; Jiang et al., 2023), aim to incorporate LLM with external knowledge to enrich its final response. A retriever is commonly used for searching related content based on keywords (Lewis et al., 2020) and is jointly fine-tuned with a sequence-to-sequence model. Despite their potential, RAG-based methods primarily address the challenges of outdated knowledge rather than resolving specific, ambiguous user queries. Unlike them, this paper focuses on empowering LLMs to seek for the missing context by actively interacting with the user, thereby improving the the answers.

**Active learning.** Active learning (Huang et al., 2010; Cohn et al., 1996) is a machine learning approach that seeks for an effective sampling strategy to select the most valuable examples and inquire an oracle for their labels. The objective of active learning is to maximize model performance while minimizing cost for oracle annotation. Existing approaches can be divided into two main categories: the informativeness-based methods and the representativeness-based methods. The former group samples examples close to the decision boundary to reduce model uncertainty (You et al., 2014; Yan & Huang, 2018). (Huang et al., 2016; Zhang et al., 2017) explored a variation for neural networks using gradient information as the metric of informativeness in text classification tasks. At the same time, the representativeness-based methods constrain the chosen data points to be distinct from each other or conform them to the data distribution (Roy & McCallum, 2001; Li et al., 2020). N-gram or word counts can be regarded as a measure of density distribution and standards for sampling practical examples (Ambati, 2012; Zhang & Plank, 2021). Preference for instances with more unseen n-grams (Erdmann et al., 2019) is another approach to selecting more representative samples. Although initially designed for supervised learning, active learning has shown to be able to assist LLMs in selecting in-context demonstrations (Margatina et al., 2023). In this work, LaMSeI utilizes active learning as a tool to select the most informative clarifying questions to pose to the user.

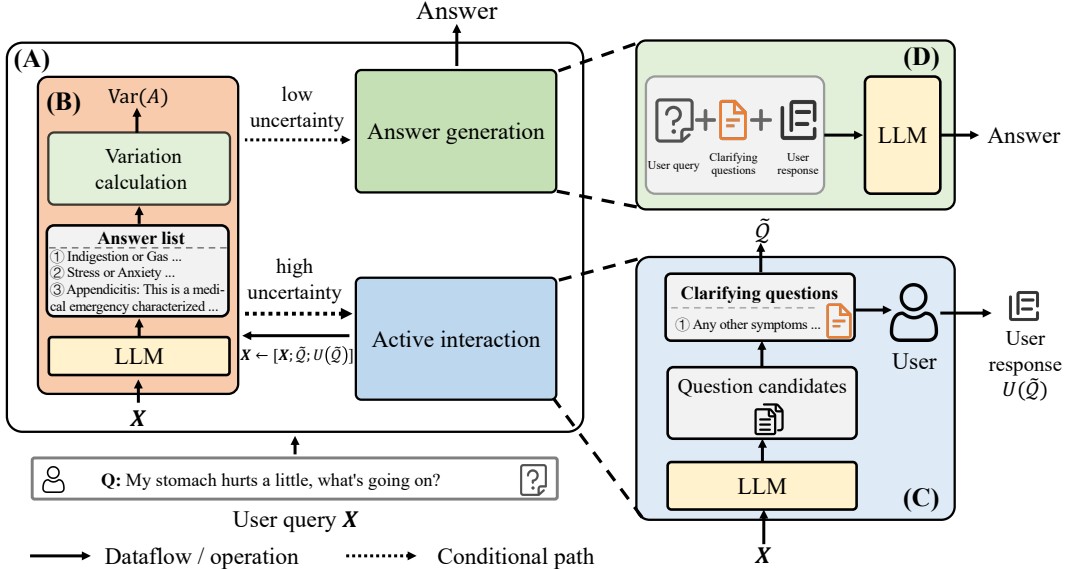

Figure 2: Illustration of LaMSeI method. **(A)** Overall workflow: the user query $X$ is processed by the uncertainty estimation module. Conditioned on the estimation result, LLM interact the user or directly generates the response. **(B)** Uncertainty estimation module evaluates the LLM's level of uncertainty regarding a query $X$. **(C)** In the active interaction module, LLM actively interacts the user by posing clarifying questions. After receiving the user's feedback, LaMSeI updates the user query to incorporate this new information and re-estimates the uncertainty. **(D)** The answer generation module generates the answer to a user query and supplementary user response (if there is user-interaction).

**Uncertainty estimation.** Uncertainty estimation (Lakshminarayanan et al., 2017; Xiong et al., 2023; Wang et al., 2025; Lewis, 1995; Loquercio et al., 2020; Malinin & Gales, 2018) is an important area of research, which measures how confident a neural network is about the predictions it makes. It distinguishes between two main types of uncertainty: aleatoric and epistemic (Abdar et al., 2021; Blundell et al., 2015). Aleatoric uncertainty arises from the inherent noise and randomness present in the data itself, which cannot be reduced even with more data. On the other hand, epistemic uncertainty is due to the model's lack of knowledge and can be potentially reduced as the model learns from more data or through improved model architectures. In this paper, we focus on the later uncertainty, as LLM's uncertainty comes from its lack of user context. To estimate these uncertainties, various techniques have been developed. Bayesian Neural Networks (BNNs (Goan & Fookes, 2020)) offer a principled statistical framework by placing priors over the model weights, thus allowing for direct estimation of uncertainty. Monte Carlo Dropout (Gal & Ghahramani, 2016) is a practical method that approximates Bayesian inference by using dropout at test time to sample from the weight distributions and thus capture uncertainty. Ensemble methods (Lakshminarayanan et al., 2017), where multiple models are trained and their predictions are aggregated, can also be used to gauge the variability in predictions as a proxy for uncertainty. However, the training of multiple models can be infeasible in the context of constrained computational resources or when constrained to a black-box model. An alternative approach is to generate multiple samples of model outputs and assess the extent of their divergence (Chen & Mueller, 2024). In our study, LaMSeI estimates LLM uncertainty by sampling a multitude of responses, with a particular emphasis on the semantic divergence of these responses.

## 3 Method

This section introduces our primary contribution, Language Model Selective Interaction (LaMSeI), which equips LLM with the ability to interact with the user for uncovering the missing user context. We describe the problem formulation (Sec. 3.1), the details of LaMSeI (Sec. 3.2), and practical implementation of LaMSeI (Sec. 3.3) in the following subsections.

### 3.1 Problem Formulation

We consider a common scenario where the user is having a conversation with the LLM $\mathcal{M}$, which takes the user query $\boldsymbol{X}$ as input and outputs the answer based on the user query: $\boldsymbol{Y} = \mathcal{M}(\boldsymbol{X})$. We expect LLM to generate a helpful response to the user query.

However, the user query may unintentionally leaves out some critical context/information that LLM cannot access, resulting in the LLM generating less helpful answers. To complete the missed piece of context, LLM actively interacts with the user with a set of clarifying questions $\tilde{\mathcal{Q}}$. Then, the user will provide the feedback $U(\tilde{\mathcal{Q}})$ to the questions as a supplemental clarification to the initial user query. The interaction above can be repeated several times to ensure that the LLM accurately grasps the user's intent and obtains enough context to answer the query. Finally, the LLM outputs the answer based on the integrated original user query and interaction content.

### 3.2 Language Model Selective Interaction

The overall workflow of LaMSeI is presented in Fig. 2. Given a user query $\boldsymbol{X}$, LaMSeI evaluates LLM's uncertainty regarding the query. If the uncertainty is high, LaMSeI inquires the user to clarify the ambiguity in the initial user query and then augments the user query with user feedback as additional information. Then, LaMSeI re-estimates the uncertainty regarding the updated query. This process is repeated until the LLM's uncertainty about user query is satisfactorily reduced. LaMSeI consists of three key components: (1) *uncertainty estimation* that evaluates LLM's uncertainty about a user query $\boldsymbol{X}$, (2) *active interaction* that asks user clarifying questions to uncover the missed context, and (3) *answer generation* that generates the answer based on the integrated information. We will elaborate on these three components in the following.

#### 3.2.1 Uncertainty Estimation by Multiple Answers Sampling

The uncertainty estimation module evaluates LLM's uncertainty about a user query $\boldsymbol{X}$ and determines whether to interact with the user actively. The previous study verifies that the model's uncertainty to an input can be estimated from an ensemble of the model outputs (Lakshminarayanan et al., 2017; Thurin et al., 2019). In our setting, the model output is in the form of text. The uncertainty can be reflected by the semantic consistency of multiple answers. To achieve this, LaMSeI samples multiple answers $\{A_1, A_2, \cdots, A_T\}$, where each answer $A_i = \mathcal{M}(\boldsymbol{X})$ is generated by the LLM under appropriate temperature parameters. To estimate the uncertainty, LaMSeI applies an embedding model to convert the array of answers into corresponding text embeddings $\{E_1, E_2, \cdots, E_T\}$. Here, we use a embedding model rather than LLM embeddings because these third-party embedding models are often optimized for general-purpose semantic similarity tasks and have been extensively tested for their ability to capture nuanced relationships between texts, which is critical for accurately estimating the uncertainty of our LLM's responses. A high variance in these embeddings indicates greater uncertainty by the LLM regarding the query, whereas a low variance suggests confidence. The variation of the answer is calculated as follows:

$$\mathrm{Var}(A) = \frac{1}{K} \sum_{k=1}^{K} \left( \frac{1}{T-1} \sum_{i=1}^{T} (E_i^k - \bar{E}^k)^2 \right), \tag{1}$$

where $K$ is the dimensionality of the embeddings, $T$ is the number of sampled responses, $E_i^k$ denotes the value of the $i$-th answer for the $k$-th dimension, and $\bar{E}^k = \frac{1}{T} \sum_{i=1}^{T} E_i^k$ is the mean value for the $k$-th dimension across all $T$ embeddings.

Subsequently, LaMSeI operates on distinct pathways, conditioned on the uncertainty estimation outcome. If the uncertainty is low, i.e., $\mathrm{Var}(A) \leq \delta$ ($\delta$ denotes the active interaction threshold), LaMSeI outputs the answer $A = \mathcal{M}(\boldsymbol{X})$. Otherwise, in cases of higher uncertainty, it indicates that LLM requires more clarification from the user for a better understanding of the user query. It should be noted that low variance does not guarantee absolute correctness, as uncertainty may still exist in certain cases. The paper uses the low variance threshold as a heuristic to guide the LLM to interact with users only when necessary, thereby improving the efficiency and reliability of the model's responses.

### 3.2.2 Active interaction with Selective Clarifying Questions

Active Interaction Module (AIM) enables LLM to interact with the user, ask clarifying questions, and better understand the user's intent. The primary focus is on how to formulate useful clarifying questions. A direct yet effective approach is leveraging LLM to generate the questions (Arora et al., 2023). Specifically, AIM prompts the LLM to generate a set of questions $\mathcal{Q}$ regarding the user's input $\boldsymbol{X}$. The specific prompts used in the experiments are presented in Appendix C. Nevertheless, questions generated this way may not be uniformly helpful and could increase the user's burden by raising too many questions simultaneously. To mitigate this, it is essential to filter and present only the most informative questions to users. To achieve this goal, we actively select most useful clarifying questions to enhance the LLM's understanding of user intent. Given a set of $N$ potential questions $\mathcal{Q}$, the goal is to select a subset of $M$ ($M \leq N$) clarifying questions $\tilde{\mathcal{Q}} = \mathcal{S}(\mathcal{Q})$. We explore two prevalent active learning strategies for this selection process, denoted as $\mathcal{S}$.

1. **Similarity-based sampling** strategy focuses on identifying questions that closely align with the user query. To achieve this, LaMSeI first extracts all questions and user query embeddings. Cosine similarity is then employed to measure the degree of semantic correspondence among embeddings, allowing for selecting the top $M$ questions that exhibit the greatest relevance to the user's query. The underlying rationale behind this strategy is the intuition that a higher degree of similarity correlates with a richer provision of pertinent information to address the user's query.

2. **Diversity-based sampling** strategy aims to capture a broad spectrum of questions. This is accomplished by encoding all questions within the set $\mathcal{Q}$ using an embedding model. Following this, the K-Means clustering algorithm (MacQueen et al., 1967) is applied to categorize the question embeddings into $M$ clusters. From each cluster, one question is chosen at random. This method ensures that the selected questions represent a diverse and extensive range of information, which can enhance the comprehensiveness of the questions gathered.

The system presents the user with this set of clarifying questions $\tilde{\mathcal{Q}}$. The user responds to these questions successively, providing clarification, denoted as $U(\tilde{\mathcal{Q}})$. Following this, the original user query is augmented to include the user's clarifications, resulting in an updated query: $\boldsymbol{X} \leftarrow [\boldsymbol{X}; \tilde{\mathcal{Q}}, U(\tilde{\mathcal{Q}})]$, which we call it *interaction-augmented query*. This augmentation draws upon the directional stimulus prompt technique (Li et al., 2023b), which has been shown to improve the model generation. Subsequently, LaMSeI re-estimates LLM's uncertainty regarding the updated user query. This uncertainty estimation and active interaction process continues iteratively until the uncertainty falls below the active interaction threshold $\delta$ or the maximum number of iterations is reached.

### 3.2.3 Answer Generation with Interaction-augmented Query

With the interaction-augmented query induced by AIM, the LLM can generate the answer to this query enriched with the additional context of user clarifications. Note that LaMSeI does necessarily inquire the user if the LLM's uncertainty about the initial user query has already been low. For clarity, the answer generation process is described here and in Fig. 2 (D) under the condition that user interaction occurs.

### 3.3 Practical Implementation

In the design of LaMSeI, we present an iterative interaction between the user and the LLM. While this iterative process can be laborious and time-consuming due to the requirement for the user to respond to clarifying questions, our practical implementation streamlines this to a single user interaction iteration. This refinement is sufficient to elucidate the user's initial query. We present this single-iteration implementation process in Algorithm 1, detailed in Appendix B. We apply `text-embedding-ada-002` model (Greene et al., 2022) to convert text to embeddings.

Table 1: Experiment results on various Q&A datasets. LaMSeI outperforms oracle method that takes supporting facts as input on certain datasets. Acc stands for the accuracy evaluated by ChatGPT. Excluding Oracle's results, the (nearly) best results are highlighted in **bold**.

| Dataset / Method | HotpotQA | | | StrategyQA | | | 2WikiMultiHopQA | | | MuSiQue | | | IIRC | | |
|---|---|---|---|---|---|---|---|---|---|---|---|---|---|---|---|
| | EM | F1 | Acc | EM | F1 | Acc | EM | F1 | Acc | EM | F1 | Acc | EM | F1 | Acc |
| DG | 29.1 | 38.3 | 44.4 | 57.2 | 57.6 | 57.6 | 17.3 | 22.9 | 43.3 | 3.8 | 14.0 | 20.5 | 14.7 | 18.1 | 20.8 |
| CoT | 32.6 | 41.6 | 48.1 | 66.7 | 66.9 | 66.9 | 31.0 | 32.9 | 49.6 | 5.5 | 12.8 | 19.8 | 17.5 | 22.0 | 24.6 |
| Self-ask | 27.7 | 38.0 | 58.6 | 34.3 | 63.2 | 34.3 | 42.5 | 49.2 | 54.0 | 15.0 | 27.0 | 28.5 | 10.1 | 30.1 | 16.8 |
| Self-ask (web) | 16.2 | 25.4 | 47.9 | 33.1 | 63.3 | 33.1 | 28.7 | 36.9 | 51.1 | 10.0 | 20.2 | 27.8 | 6.0 | 10.7 | 25.6 |
| CRUD-RAG | 29.0 | 37.6 | 40.5 | 37.0 | 45.3 | 48.0 | 6.5 | 16.3 | 43.0 | 12.0 | 23.1 | 27.5 | 25.0 | 33.1 | 35.0 |
| Oracle context | 59.4 | 86.9 | 72.4 | 78.2 | 78.2 | 78.1 | 60.5 | 72.0 | 83.5 | 21.0 | 33.5 | 36.0 | 29.4 | 62.5 | 42.6 |
| LaMSeI | 47.5 | **58.7** | **68.1** | 66.3 | 66.4 | 66.4 | 42.8 | 52.0 | **71.3** | 16.5 | 25.9 | **30.5** | **34.1** | **42.6** | **51.1** |
| LaMSeI+CoT | **49.1** | **59.2** | **69.6** | **71.7** | **71.7** | **71.7** | **49.8** | **61.1** | **73.0** | **18.5** | **27.7** | **31.5** | 27.2 | 36.6 | 45.1 |

# 4 Experiment

In this section, we evaluate the efficacy of LaMSeI through extensive experiments on diverse, challenging datasets. The objective is to validate LaMSeI's capacity to improve LLM's understanding of user queries and to improve the quality of the responses provided. We aim to answer the following essential questions: (1) How does LaMSeI perform in comparison to current answer generation methods across different themes of user query (Sec. 4.2,4.5)? (2) Can LaMSeI gain useful information from the selective interaction (Sec. 4.3)? (3) Can LaMSeI be integrated with various LLMs (Sec. 4.4)? and (4) What is the impact of each component and parameter on the performance of LaMSeI (Sec. 4.7)? We begin with introducing the experimental setting.

## 4.1 Experimental Setup

In this subsection, we first introduce how to construct user interaction framework to conduct our experiments. Then, we detail the datasets used in the experiments, followed by the introduction of the baseline methods. Finally, we present implementation details and the evaluation settings.

**Setup for LLM-user interaction.** Our experiments require a procedure wherein an user interacts with the LLM. An ideal setup would involve a human user engaging with the LLM, responding to its clarifying questions. However, this setup is impractical due to the high costs and labor-intensive nature of human participation. Follow existing work (Li et al., 2024), we employ GPT-4 as a proxy for human user in the majority of our experiments. We use datasets in which each problem comprises a query paired with supporting facts that serve as a context for the query. The nature of these datasets is particularly suited for our experiments, as they are designed to simulate the absence of user-provided context. Note that to ensure a comprehensive evaluation, we also conduct experiments with actual human participation, which are described in Sec. 4.5.

**Dataset.** We conduct experiments on five challenging Q&A datasets alongside a dataset dedicated to meeting summarization. Tab. 2 shows some examples from these datasets. In our experiments, all methods, if not specifically marked, are agnostic to the supporting facts. These datasets are: (1) **HotpotQA** (Yang et al., 2018) is collected on the English Wikipedia. Each question in the dataset comes with the two gold paragraphs, as well as a list of sentences that crowdworkers identify as supporting facts necessary to answer the question; (2) **StrategyQA** (Geva et al., 2021) consists of questions similar to that in HotpotQA, but the answer format is limited to True or False; (3) **2WikiMultiHopQA** (Ho et al., 2020) uses structured and unstructured data and introduces the evidence information containing a reasoning path for multi-hop questions; (4) **MuSiQue** (Trivedi et al., 2022) is a multi-hop QA dataset with 2-4 hop questions using seed questions from five single-hop datasets; (5) **IIRC** (Ferguson et al., 2020) is a dataset for incomplete information reading comprehension, providing only partial information to answer them, with the missing information occurring in one or more linked documents; (6) **QMSum** (Zhong et al., 2021) contains dialogue histories of multi-domain meeting. The user queries LLM to answer specific questions about the meeting, while some parts of the dialogue are masked.

Table 2: Examples of user query, supporting facts, and correct answer for the datasets used in the experiments.

|  | HotpotQA | StrategyQA |
|---|---|---|
| User query | Were Up and The Watercolor released in the same year? | Are more people today related to Genghis Khan than Julius Caesar? |
| Supporting facts | Up and The Watercolor are two films. Up was released in ... | Compare the number of their offspring. Julius Caesar had three children. Genghis Khan had sixteen children ... |
| Label | Yes | True |

**Baseline.** We compare LaMSeI with five representative baseline methods that are widely applied in the community. (1) Direct Generation (**DG**) directly generates the answer with the deterministic output of the LLM; (2) Greedy asking (**GA**) is a baseline that greedily asks the user N clarifying questions. It directly prompts the LLM to generate N clarifying questions to pose to the user. (2) Chain of Thought (**CoT**) (Wei et al., 2022) generates the answer by prompting the LLM to reason the results through a series of intermediate reasoning steps. We use the '*Let's think step by step*' prompt with few-shot demonstrations in the experiments; (3) **Self-ask** (Press et al., 2023) poses and responds to the self-generated follow-up questions, refining its understanding before providing the initial query's response; (4) **Self-ask (Web)** extends the Self-ask method by integrating a web-search API, specifically the Bing search[1], to incorporate external knowledge into its answering process; (5) **CRUD-RAG** (Lyu et al., 2024) uses RAG to retrieve relevant paragraphs or sentences from vector databases. Here, we use `all-mpnet-base-v2` model from `sentence-transformers` library (Reimers & Gurevych, 2019) to extract embeddings and compute cosine similarity. (7) **Oracle** generates the answer directly but is provided ground-truth supporting facts.

**Evaluation and Metrics.** We evaluate various methods on the first 400 questions from the training set across five Q&A datasets. To ensure a fair comparison, we employ in-context demonstration for all methods, selecting two random examples from each dataset to serve as demonstrations. In this way, the LLMs are able to generate answers in a standardized format. We evaluate performance using exact match (EM) and F1 scores across the Q&A datasets. Following Iter-RetGen (Shao et al., 2023), we also evaluate answer accuracy (Acc) using the `gpt-3.5-turbo` model, as direct calculation of answer accuracy is not feasible on these datasets. For the QMSum dataset, which contains long text, we apply `gpt-4` to evaluate the answer preference over the DG method.

**Implementation Details.** We utilize ChatGPT (OpenAI, 2023) as the primary model for our experiments, a popular model in the field. To reduce human costs in providing feedback to clarifying questions, we deploy ChatGPT in dual capacities: as an LLM (GPT-3.5) and as a pseudo-human interlocutor (GPT-4) in the experiment. We present the queries from the dataset to the LLM, while the supporting facts are provided to only the GPT-4 model. In this way, the user query needs more context and information to be answered. To ensure the experiment's integrity, we restrict GPT-4 from accessing the original user questions when it answers the clarifying questions. We also consider LaMSeI+CoT method, which equips LaMSeI with *Let's think step by step* prompt and incorporate demonstrations to provide a fair and competitive comparison. The experiments are conducted with $2 \times$ NVIDIA 3090 and AMD EPYC 9654 96-Core processor. More detailed information regarding the experimental setting, such as prompts and parameters, is available in Appendix C.

### 4.2 Main Results

**Performance on Queries with Incomplete Context.** Tab. 1 presents the comparative performance of LaMSeI against baseline methods across five Q&A datasets. Overall, LaMSeI consistently surpasses the baseline methods, achieving an average answer accuracy of 50.9%, which marks a significant improvement compared to the basic Q&A framework DG (31.9%). This result underscores LaMSeI's superiority in improving model's answer on queries with incomplete context. The CoT method, specifically designed to enhance the reasoning capabilities of LLM, also falls short of LaMSeI's performance. The limitation of CoT

---

[1]Bing search API is accessible at `https://www.microsoft.com/en-us/bing/apis/bing-web-search-api`.

for our setting lies in that it completely relies on the LLM's embedded knowledge, which does not actively seek out additional context that may be absent in the user's query. Similarly, the Self-ask strategy, which also leverages the LLM's embedded knowledge, cannot address the gaps in information presented by the user, resulting in a performance disparity with LaMSeI. Self-ask (web) is a typical RAG-based method that seeks information from the web. The experiment results of Self-ask (web) suggest that the external knowledge base does not demonstrate an advantage in dealing with ambiguous user queries, as indicated by its close performance to the DG method. CRUD-RAG is another RAG-based method that builds a vector database and leverages the retriever to retrieve relevant content for specific queries. The result reveals that directly retrieving related information and then using it as context information is inferior to LaMSeI 's iterative form. Notably, LaMSeI achieves comparable performance with the Oracle method on HotpotQA and StrategyQA and outperforms it on IIRC. These results provide strong evidence that LaMSeI effectively acquires valuable information from the user, improves the model's comprehension of user queries, and leads to improved performance.

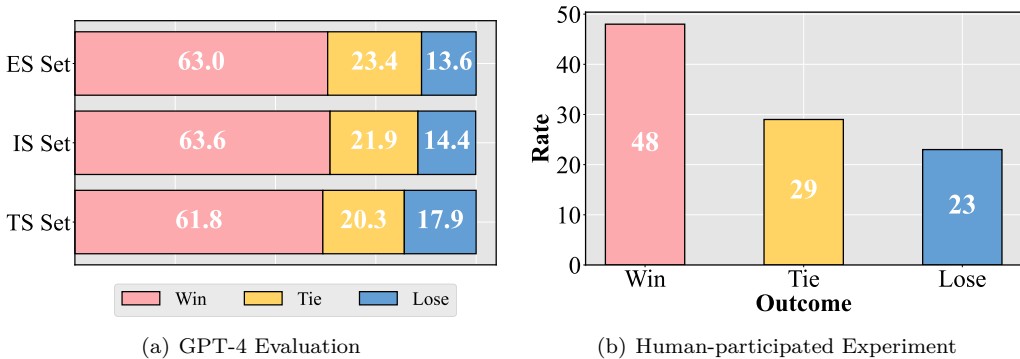

(a) GPT-4 Evaluation          (b) Human-participated Experiment

Figure 3: Comparative and analytical results for LaMSeI on the QMSum dataset: **(a)** GPT-4-based evaluation showing LaMSeI's win rate against DG on three subsets of QMSum. **(b)** Human-evaluated win rate of LaMSeI output against DG. In this experiment, LaMSeI interacts with a human participant, who accesses to the supporting facts.

**GPT-4 evaluation on QMSum.** We conduct experiments on the QMSum dataset, wherein the language model is asked to summarize specific perspectives based on the meeting's dialogue history. Assessing the quality of the summaries generated by the model is a significant challenge, as traditional reference-based metrics like BLEU and ROUGE often correlate poorly with human judgment. In our study, we employ GPT-4 as a reference-free metric for evaluation, which has demonstrated a high degree of alignment with human assessments (Liu et al., 2023). Fig. 3(a) illustrates that LaMSeI surpasses the DG method by a margin exceeding 60% in win rate and 20% in tie rate. These results underscore the efficacy of LaMSeI, which, through active interaction, garners valuable information from users to accurately respond to queries, thereby demonstrating the effectiveness of our approach.

## 4.3 Performance of Selective Interaction

**Comparison with greedy asking.** To verify whether the selective interaction in LaMSeI effectively selects meaningful questions to interact with the user, we conduct experiments on QMSum to compare LaMSeI with greedy asking (GA) method that greedily interacts with users with five clarifying questions by prompts LLM to generate the questions. The results are presented in Tab. 3, where each score represents the method's winning rate over direct generation method. There are two main observations: (1) LaMSeI outperforms GA method, although asking only three clarifying questions during the interaction, as opposed to GA's five questions. This results demonstrate that LaMSeI can pose effective clarifying questions through selective interaction and active learning question selection. (2) All these interactive-LLM method improve performance over DG method, achieving more than 60% winning rate over direct answer generation. This results highlight the importance of interacting with users to seek for more context.

Table 3: Comparison of LaMSeI with greedy asking. GA method greedily asks five clarifying questions before answering, while LaMSeI asks only three questions when interacting. The score in the table represents the method's winning rate over DG method.

|  | GA | LaMSeI (Diversity) | LaMSeI (Similarity) |
|---|---|---|---|
| Score | 63.04 | 63.04 | 65.76 |

**Analysis on model outputs.** Previous results have demonstrated LaMSeI enhances the capability of LLM to generate improved responses. To investigate the source of this enhancement and its influence on the LLM's understanding on user queries, we conduct a case study analyzing the running process of LaMSeI method, which covers successful case and failed cases, as shown in Tab. 9 in Appendix. The analysis yields several critical insights. Firstly, LaMSeI can raise effective clarifying questions to seek clarification for the user query. For instance, $\tilde{\mathcal{Q}}_1$ in the success case, LaMSeI inquires about the specific actions that lead to injuries, aligning precisely with the supporting facts. Secondly, applying active learning strategies filters the most informative questions from the question candidates. Overall, the quality of clarifying questions directly determines whether LaMSeI successfully resolves ambiguity and generates accurate answers. In the success case, the selected questions (e.g., "Are there specific movements in waltz/slam dancing that could lead to injuries?") precisely target the core of the user query ("Is waltz less injurious?"). These questions align with the supporting facts (e.g., slam dance involves "collisions"), enabling LaMSel to extract critical context about injury mechanisms and produce a correct answer. Conversely, in the failure case, the selected questions (e.g., "Other income sources for Billie Eilish?" or "Average Porsche price?") are misaligned with the query's key uncertainty ("Can she afford it?"). They overlook the decisive evidence in the supporting facts, leading to an incorrect answer despite sufficient information being present. This contrast confirms that high-quality questions must directly address the information gap implied by the query and supporting context; irrelevant or overly broad questions fail to reduce uncertainty and degrade answer accuracy. Future work could better leverage this feature to generate clarify questions.

Table 4: Experiments on various open-sourced LLMs with/without LaMSeI technique. LaMSeI consistently improves the performance of different LLM backbones.

| Dataset / Method | HotpotQA | | StrategyQA | | 2Wiki* | | MuSiQue | | IIRC | |
|---|---|---|---|---|---|---|---|---|---|---|
|  | EM | Acc | EM | Acc | EM | Acc | EM | Acc | EM | Acc |
| LLaMA2-7B-DG | 5.5 | 25.3 | 50.3 | 60.6 | 15.2 | 37.0 | 1.3 | 13.7 | 1.5 | 15.3 |
| LLaMA2-7B-CoT | 15.8 | 28.3 | **56.1** | **67.4** | 20.0 | 29.3 | 2.0 | 14.0 | 7.0 | 16.3 |
| LLaMA2-7B-LaMSeI | 8.7 | 19.8 | 41.6 | 43.6 | 25.6 | **69.7** | 3.0 | 23.5 | 12.0 | 21.3 |
| LLaMA2-7B-LaMSeI-CoT | **21.3** | **33.6** | 51.4 | 51.1 | **39.7** | 62.0 | **10.7** | **32.5** | **12.3** | **22.3** |
| Vicuna-7B-DG | 2.0 | 19.0 | 50.5 | 47.6 | 8.1 | 8.7 | 1.0 | 7.0 | 7.4 | 6.5 |
| Vicuna-7B-CoT | 12.5 | 29.1 | 56.6 | 58.6 | 12.0 | 13.5 | 1.5 | 10.0 | 9.2 | 17.5 |
| Vicuna-7B-LaMSeI | 27.3 | 55.9 | 57.6 | 59.1 | 18.7 | 28.2 | 7.7 | 24.0 | 19.3 | 35.1 |
| Vicuna-7B-LaMSeI-CoT | **29.1** | **56.9** | **62.4** | **63.9** | **33.2** | **31.0** | **7.9** | **25.2** | **21.5** | **38.1** |
| LLaMA3-8B-DG | 15.9 | 22.0 | 65.5 | 64.0 | 18.3 | 36.5 | 3.0 | 16.7 | 7.8 | 15.4 |
| LLaMA3-8B-CoT | 17.8 | 40.8 | 59.0 | 59.1 | 29.6 | 36.2 | 4.1 | 17.7 | 13.1 | 18.5 |
| LLaMA3-8B-LaMSeI | 50.5 | 66.4 | 62.1 | 61.2 | 49.5 | 58.6 | 11.0 | 32.0 | 19.3 | 25.4 |
| LLaMA3-8B-LaMSeI-CoT | **52.6** | **67.2** | **79.5** | **76.0** | **51.3** | **65.1** | **15.2** | **32.7** | **31.6** | **40.1** |
| Vicuna-13B-DG | 12.0 | 36.8 | 56.8 | 57.8 | 10.0 | 32.3 | 9.0 | 15.0 | 3.5 | 9.5 |
| Vicuna-13B-CoT | 20.0 | 38.1 | 60.3 | 60.9 | 30.8 | 40.5 | 11.5 | 17.2 | 5.5 | 11.1 |
| Vicuna-13B-LaMSeI | **34.8** | 60.1 | 55.8 | 56.1 | 12.3 | 58.5 | 10.0 | **33.0** | 17.0 | 24.1 |
| Vicuna-13B-LaMSeI-CoT | 33.8 | **60.6** | **65.6** | **67.1** | **32.5** | **63.3** | **12.0** | 30.2 | **18.8** | **24.2** |

## 4.4 Applicability on Different LLMs

We verify the applicability of LaMSeI by assessing its performance on different LLMs. We conduct experiments with LLaMA2-7B (Touvron et al., 2023), Vicuna-7B (Zheng et al., 2023b), LLaMA3-8B (Team, 2024a) and

Vicuna-13B. These models are all from the open-sourced version. The experimental setup is consistent with the main experiments, with the results shown in Tab. 4. The results demonstrate that LaMSeI surpasses the DG and CoT methods on two datasets. This suggests that LaMSeI is well-suited to models with fewer parameters. Besides, LaMSeI can improve the performance of a wide range of LLMs, justifying the applicability of the proposed method.

## 4.5 Experiments with Human Participant

In addition to experiments employing GPT-4 as a simulated human interlocutor, we also conduct human-participated experiments where actual human interaction is integrated. The experiment is conducted using the QMSum dataset, from which 100 user queries are randomly selected for evaluation. We invite five participants to the experiment, each accounting for 20 queries. Participants respond to clarifying questions posed by the LLM, which subsequently generates answers informed by the human response. Subsequently, another participant compares these anonymous answers generated by LaMSeI and DG and annotates a preference for one. The experimental results are illustrated in Fig. 3(b). The human evaluation results reveal that LaMSeI outperforms or is comparable to DG in 77% of the instances. This result demonstrates LaMSeI's superior ability to comprehend user input during human interaction and verifies the feasibility of applying LaMSeI in real application.

Table 5: Performance of LaMSeI on user query with different levels of context masking. We use GPT-4 with evaluation prompt (Fig. 6) to assess the response quality of LaMSeI against DG.

| Mask Rate | ES | | | IS | | | TS | | |
|---|---|---|---|---|---|---|---|---|---|
| | Win | Lose | Tie | Win | Lose | Tie | Win | Lose | Tie |
| 0 | **47.86** | 34.24 | 17.90 | **48.13** | 36.90 | 14.97 | **55.28** | 30.08 | 14.63 |
| 0.3 | **55.14** | 23.67 | 21.22 | **58.70** | 31.52 | 9.78 | **48.96** | 34.38 | 16.67 |
| 0.5 | **63.04** | 23.35 | 13.62 | **63.64** | 21.93 | 14.44 | **61.79** | 20.33 | 17.89 |
| 0.7 | **66.17** | 21.05 | 20.54 | **71.20** | 13.91 | 15.71 | **62.12** | 25.00 | 12.88 |

## 4.6 User Query with Less Context

LaMSeI actively seeks clarification from users to gain additional context regarding their queries. To evaluate this capability for context-seeking, we conduct experiments on the QMSum dataset, with varying masking ratio applied to the dialogue history. We implement the context mask by directly masking the tokens of support facts that are exposed to the LLM. As the masking rate increases, the availability of contextual information to the LLM correspondingly decreases, allowing LaMSeI to demonstrate stronger relative gains by actively seeking missing information. As the experimental results are shown in Tab. 5, when larger proportions of context are masked, LaMSeI demonstrate more significant advantage over the DG method. This improved performance is attributed to LaMSeI's ability to seek useful context from the user, which mitigates the challenges of insufficient contextual data. Furthermore, it is observed that LaMSeI maintains its efficacy in eliciting additional information to answer the user query, even when the full context is available, thereby enriching its responses to user queries.

## 4.7 Ablation Study

In this subsection, we conduct an ablation study to investigate the influence of different components of LaMSeI on the algorithm performance, as the results presented in Fig. 4.

**Number of clarifying questions.** The number of clarifying questions posed to the user, denoted as $M$, is an essential parameter in LaMSeI framework. These ablation results are presented in Fig. 4. Overall, there is a clear improvement in LaMSeI's performance correlating with an increase in the number of clarifying questions $M$. This trend substantiates the efficacy of LaMSeI's clarifying questions in mitigating the contextual gap between the LLM and the user. We observe that $M = 3$ is sufficient to achieve notable performance gains. This performance improvement occurs because the LLMs can pose more questions, gathering additional

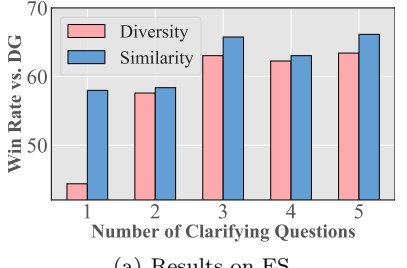 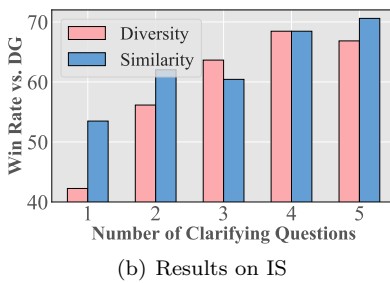 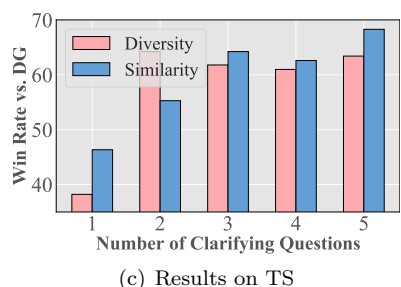

|  (a) Results on ES | (b) Results on IS | (c) Results on TS |

Figure 4: Ablation study on the impact of the number of clarifying questions and active learning strategies on (a) ES dataset, (b) IS dataset, and (c) TS dataset. The values in the figure represent the win rate of LaMSeI against DG method.

pertinent context to address the user's query. While larger $M$ potentially contributes to superior performance, it also imposes a heavier demand on the user to provide feedback. Therefore, it is essential to balance user experience with the quality of the model's responses. Thus, we recommend adopting a moderate value of $M$, such as 3, for practical applications of LaMSeI.

**Active Learning Selection Strategy.** LaMSeI employs active learning strategy to select informative questions to inquire the user. We present the experimental results of LaMSeI using distinct active learning selection strategies, as illustrated in Fig. 3. Overall, the similarity strategy, which prioritizes selecting clarifying questions that are semantically similar to the user's query, outperforms the diversity strategies. These findings suggest that a focus on diversity may not be as effective in sifting out the most informative questions, and that the similarity of the questions to the user's query is a more critical factor.

**Active Interaction threshold.** In LaMSeI, the active interaction threshold, denoted as $\delta$ is important in determining the instances when the LLM actively interacts with the user. To evaluate the impact of $\delta$ on LaMSeI, we conduct experiments with different values of $\delta$. The results in Tab. 6 indicate an enhanced performance of LaMSeI at lower $\delta$ values, which correlates with a higher propensity for actively inquiring users. Notably, at a more significant threshold, e.g., $\delta = 0.015$, the performance of LaMSeI approximates that of the DG method, attributed to the reduced frequency of active interactions. This observation encourages us to set a relatively low $\delta$ value to enable active interaction users by the LLM.

| $\delta$ | ES | | | IS | | | TS | | |
|---|---|---|---|---|---|---|---|---|---|
| | **Win** | **Lose** | **Tie** | **Win** | **Lose** | **Tie** | **Win** | **Lose** | **Tie** |
| 0.005 | **49.42** | 15.56 | 35.02 | **42.78** | 16.57 | 40.65 | **40.65** | 12.20 | 47.15 |
| 0.010 | **23.57** | 10.72 | 65.71 | **20.32** | 6.95 | 72.73 | **19.51** | 5.69 | 74.80 |
| 0.015 | **11.28** | 2.33 | 86.39 | **10.16** | 3.74 | 86.10 | **8.94** | 4.87 | 86.19 |

Table 6: Ablation study on the effect of the active inquiry threshold on the QMSum dataset. We use GPT-4 with evaluation prompt (Fig. 6) to assess the response quality of LaMSeI against DG.

## 5   Conclusion and Limitation

This paper highlights the need for LLMs to be interactive to better understand user's intent. To achieve this, we propose a novel LaMSeI method, which empowers LLMs to selectively interact with users, effectively handling user queries with incomplete context. Comprehensive experiments demonstrate that LaMSeI clearly improves LLM's grasp of user intent, uncovers the missed context and leads to more reliable model response. Despite the superior performance, LaMSeI has certain limitations. First, LaMSeI straightforwardly prompts LLM to generate a set of question candidates. As shown in our case study in Sec. 4.3, this process may only sometimes yield sufficient and informative questions even after the active learning process. As a potential improvement, there has been a learning-based method (Komeili et al., 2022) that shows promising results in

generating practical clarifying questions by training a query generator. Second, the current embedding-based method for assessing the LLM's uncertainty about a user's query assumes that the embedding model is able to accurately captures the semantic information of the original text. An alternative could involve estimating the model's uncertainty based on the likelihood of the generated response. Lastly, our experiments utilize sampled questions from existing Q&A datasets to mimic user queries, which may only partially represent actual user interactions in the real world. It would be interesting to evaluate LaMSeI's performance when facing more realistic user queries and interacting with humans. We encourage future research to explore these interesting fields and develop more effective approach for interactive LLMs.

## Acknowledgement

This work is supported by the NSFC (U2441285, 62222605). The authors extend their appreciation to anonymous reviewers for providing valuable comments during the peer-review process.

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

## A    Discussion

### A.1    Usage of Datasets in Experiments

In the current research, we systematically conduct experiments using a range of well-established datasets to assess the performance of LaMSeI. These experiments involve creating scenarios where user queries are ambiguous by withholding supporting facts. However, these datasets may only partially represent the complexity and nuances of real-life conversational scenarios where users pose diverse and often intricate questions. To bridge this gap and enrich the experimental outcomes, future research could incorporate questions drawn from everyday human interactions. Additionally, exploring alternative datasets encompassing more extensive chatting histories could yield more comprehensive insights. Besides, the current experimental framework primarily utilizes GPT-4 OpenAI (2023) as a simulated conversational partner, offering responses to clarification questions. While simulating real-world scenarios by providing GPT-4 access to supporting facts, this setup may only partially capture the conversational style typical of human interactions. The initial phase of our research involved a preliminary assessment of LaMSeI's competency in engaging with human participants. This provided valuable insights into its interactive capabilities. For a more robust evaluation, it would be beneficial to expand the scope of the experiments to include a more comprehensive array of human participants. Engaging these participants in conversations with LaMSeI, utilizing their unique inquiries and feedback, would offer a more authentic and varied perspective on the model's interactive performance and capabilities.

### A.2    Active Learning for LLMs

Active learning Huang et al. (2010) is acclaimed for its efficacy in augmenting model performance, particularly in scenarios where obtaining a label is laborious and costly Margatina et al. (2023). Prior research combines active learning and LLMs, mainly focusing on selecting in-context demonstrations for LLMs. Unlike them, in this study, LaMSeI leverages active learning techniques by selecting the most informative clarifying questions to inquire user. This strategy enables LLMs to learn from relevant and valuable responses. The utility of active learning extends to its capacity for tailoring interactions based on user feedback, fostering a dynamic and user-focused engagement.

Nevertheless, there are some challenges to involving active learning techniques. A primary concern is the initial selection and crafting of clarifying questions, which must be sufficiently broad to encompass the user's potential intent while remaining focused enough to steer the model toward constructive clarifications. Furthermore, there exists a challenge concerning the potential bias inherent in the queries chosen by the active learning algorithm. Such bias could inadvertently direct the model's learning trajectory in a specific,

perhaps unintended direction. Addressing this issue, future enhancements might involve the development of more advanced question-generation methods. These methods should possess a deeper understanding of the nuances and context of user queries and incorporate a more diverse array of data sources to counteract bias. Additionally, the evolution of active learning to accommodate more intricate, multi-turn interactions could improve the richness and quality of dialogues between users and the LLM. Such advancements would likely result in more sophisticated and precise responses from the LLM, thereby enhancing the overall effectiveness and user experience.

## B  Practical Algorithm of LaMSeI

In Sec. 3.2, the LaMSeI method is delineated through an iterative framework. Nonetheless, in practical applications, this iterative process of querying the user to diminish the uncertainty associated with the LLM's understanding of the user query may be time-intensive and onerous. To address these challenges, we introduce a streamlined version of LaMSeI, implemented with a singular inquiry iteration. The details of this practical implementation are outlined in Algorithm 1.

---

**Algorithm 1** Practical Implementation of LaMSeI

---

**Require:** User input $\boldsymbol{X}$, active learning selection strategy $\mathcal{S}$, active inquiry threshold $\delta$, number of clarifying questions $M$

   Sample a set of answers to user query: $\{\mathcal{A}_i = \mathcal{M}(\boldsymbol{X})\}$
   Calculate the variation of the answers $\text{Var}(A)$ (Eq.1)
   **if** $\text{Var}(A) < \delta$ **then**
     // Low uncertainty
     Generate the answer directly: $\boldsymbol{Y} = \mathcal{M}(\boldsymbol{X})$
   **else**
     // Active inquiry
     Generate a set of clarifying questions $\mathcal{Q}$
     Select questions from the set with active learning strategy: $\tilde{\mathcal{Q}} = \mathcal{S}(\mathcal{Q})$
     Inquire the user and get the feedback $U(\tilde{\mathcal{Q}})$
     Generate the answer $\boldsymbol{Y}$
   **end if**
**Return:** Answer to user query $\boldsymbol{Y}$

---

## C  More Experiment Details

This section provides more details of the experiments, including prompts, hyper-parameters, and more examples of the datasets.

### C.1  Prompts

We present the prompts used in the experiments in three topics: prompts used for LaMSeI (Fig. 5), evaluation prompts based on Zheng et al. (2023a) and prompts for baselines, which are depicted in Fig. 6, and Fig. 7, respectively.

### C.2  Hyper-parameters

Tab. 7 presents hyper-parameters used in our experiments. To implement baseline methods, we utilize their official implementation or released hyper-parameters.

---

**Prompts for LaMAI**

**(Generate clarifying questions for Q&A datasets)**
Now you need to ask some questions about the query to help complete the question better. Note that the questions you pose should be useful for assistant to answer the original user query. List your questions in order and separate them by ';'.
The user query is: {USER QUERY}. Your posed questions are:

---

**(Generate clarifying questions for QMSum)**
Here is the dialog history of a multi-turn conversation: {DIALOG HISTORY}.
User query: {USER QUERY}. Do you have any questions to address the user query?
If so, please list and separate them by ';'. Your posed questions are:

---

**(Answer clarifying questions for GPT-4)**
Supporting facts: {SUPPORTING FACTS}. User query: {USER QUERY} Please answer the following questions based on the provided supporting facts and user query. You should make full use of the supporting facts to answer the questions, and your answer must be helpful.
Questions are: {CLARIFYING QUESTIONS}

---

**(Answer generation)**
Question: {USER QUERY}. You should answer the user query based on the hints (a set of question-answer pairs).
Please answer the question directly without explanation. Give the answer following the formats of the given examples.
Example question: {EXAMPLE QUESTION}. Example answer: {EXAMPLE ANSWER}.
Hint 1: {CLARIFYING QUESTION 1}: {ANSWER 1}
Hint 2: {CLARIFYING QUESTION 2}: {ANSWER 2}
Hint 3: {CLARIFYING QUESTION 3}: {ANSWER 3}

Figure 5: Prompts for LaMSeI.

---

**General Prompts - GPT-4 Evaluation**

Please act as an impartial judge and evaluate the quality of the responses to the user task displayed below.
You should rate these two answers based on which one is closer to the ground truth answer. Your evaluation should consider factors such as the helpfulness, relevance, accuracy, depth, creativity, and level of detail of their responses. Begin your evaluation by comparing the two responses and provide a short explanation. Avoid any positional biases and ensure that the order in which the responses were presented does not influence your decision. Do not allow the length of the responses to influence your evaluation.
Please provide your explanation, and output your final verdict by strictly following this format:
Verdict: [explanations]
Choice: A, B or C.
"A" if answer A is better, "B" if answer B is better and "C" for a tie.
Here is the ground truth answer: {GROUND TRUTH ANSWER}.
Two answer candidates:
[A]: {ANSWER1} [B]: {ANSWER2}.

Figure 6: Prompt for GPT-4 evaluation.

---

**Prompts for Baseline Methods**

**(Direct Generation)**
Please answer the question directly without explanation. Just give the answer and do NOT need to give the explanation. You should follow the formats of following examples.
Question: Which film has the director born first, Captain Phantom or Brasileirinho (Film)?
Answer: Captain Phantom
Question: {TASK}

**(Direct Generation - Cot)**
You should follow the formats of following examples. First think step by step, then give your own answer:
So the answer is: [your answer].
Note that [your answer] should be short and concise, and do NOT need the explanation.

**(Direct Generation with Knowledge - Oracle)**
Here is the dialog history of a multi-turn conversation: {DIALOG HISTORY}.
User query: {USER QUERY}. Do you have any questions to address the user query?
If so, please list and separate them by ';'. Your posed questions are:

**(Self-Ask)**
Question: Who lived longer, Theodor Haecker or Harry Vaughan Watkins?
Are follow up questions needed here: Yes.
Follow up: How old was Theodor Haecker when he died?
Intermediate answer: Theodor Haecker was 65 years old when he died.
Follow up: How old was Harry Vaughan Watkins when he died?
Intermediate answer: Harry Vaughan Watkins was 69 years old when he died.
So the final answer is: Harry Vaughan Watkins.
Are follow up questions needed here: (Here, you can say 'Yes.Follow up: ...', like demonstrations mentioned above, or 'No. So the final answer is:...' to provide the answer directly)

---

Figure 7: Prompts for baseline methods.

| Name | Value |
|---|---|
| Num. of clarifying questions $M$ | 3 |
| $\delta$ | 0.005 |
| temperature for uncertainty estimation | 0.5 |
| top_p | 1 |
| presence penalty | 1 |
| sample strategy | diversity |
| Num. of demonstration | 2 |

Table 7: Hyper-parameters used in the experiments.

### C.3  Examples of Datasets

We present examples from datasets in Tab. 8. The table shows that each task consists of a user query and corresponding supporting facts or user intent. LaMSeI needs to propose the most valuable questions based on the context to seek clarification from the user to respond with a more refined answer.

| | User query | Supporting facts / User intent | Label answer |
|---|---|---|---|
| HotpotQA | Musician and satirist Allie Goertz wrote a song about the "The Simpsons" character Milhouse, who Matt Groening named after who? | "Lisa Marie Simpson is a fictional character in the animated television series, The Simpsons. She is the middle child and most intelligent of the Simpson family"... | President Richard Nixon |
| StrategyQA | Could Lil Wayne legally operate a vehicle on his own at the beginning of his career? | Lil Wayne's career began in 1995, at the age of 12, when he was signed by Birdman and joined Cash Money Records as the youngest member of the label... | False |
| 2WikiMultiHopQA | Are director of film Move (1970 Film) and director of film Méditerranée (1963 Film) from the same country? | Move is a 1970 American comedy film... and directed by Stuart Rosenberg. The screenplay was written by... | No |
| Musique | What is the highest point in the country where Bugabula is found? | Bugabula is one of the five traditional... It is located in the Kamuli District. Iran consists of the Iranian Plateau... | 1400 meters |
| IIRC | How old was Hokutoumi when he defeated Jingaku Takashi by making him stumble out of the dohyo? | He came from the same area of Japan as future stable-mates Sakahoko and Terao. He was fond of kendo at school. He joined Izutsu stable in 1977... | 27 years |

Table 8: Examples of user query, supporting facts and correct answer for the datasets used in the experiments.

Table 9: A case study on the model output for LaMSeI, with a success and a failure cases. More examples are in Appendix.

| User query | *Success case*: Is waltz less injurious than slam dance? | *Failure case*: Can Billie Eilish afford a Porsche? |
|---|---|---|
| Supporting facts | The waltz is a rhythmic dance performed in triple time by a couple. A slam dance is a type of dance in which leaping dancers collide against each other. | Billie Eilish is a famous female singer. Billie Eilish is 18 years old and has a net worth of $25 Million. A Porsche Boxster is a car that starts at $59,000. $25,000,000 is greater than $59,000. |
| Questions before selection | $\mathcal{Q}_1$: What are the potential injuries associated with slam dancing? $\mathcal{Q}_2$: Are there any specific movements or techniques in waltz that could lead to injuries? ... (10 questions in total) | $\mathcal{Q}_1$: Does Billie Eilish have any other expensive assets or investments? $\mathcal{Q}_2$: Does Billie Eilish have any endorsement deals or sponsorships that could contribute to her ability to afford a Porsche? ...(10 questions in total) |
| Questions after selection | $\tilde{\mathcal{Q}}_1$: Are there any specific movements or techniques in waltz that could lead to injuries? $\tilde{\mathcal{Q}}_2$: Are there any specific movements or techniques in slam dancing that could lead to injuries? | $\tilde{\mathcal{Q}}_1$: Are there any known sources of income for Billie Eilish besides her music career? $\tilde{\mathcal{Q}}_2$: What is the average price range for a Porsche? |
| Output / Label | True / True | False / True |

# D    Additional Results

In this section, we present additional experimental results that are omitted in the main text due to the space limitation.

## D.1    Experiments with More Models

To better verify the applicibility of LaMSeI, we conduct experiments with Qwen-2.5-7B and Qwen-2.5-14B models Team (2024b) on 2WikiMultiHopQA and MuSiQue datasets, as the results shown in Tab. 10. Overall, LaMSel can also improve these advanced models' performance, outperforming baselines such as DG and CoT, confirming LaMSeI's effectiveness on more recent models.

Table 10: Experiments on Qwen models, which are advanced models published recently.

| Dataset 
 Method | 2WikiMultiHopQA | | MuSiQue | |
|---|---|---|---|---|
| | EM | Acc | EM | Acc |
| Qwen2.5-7B-DG | 17.5 | 21.5 | 3.0 | 9.6 |
| Qwen2.5-7B-CoT | 6.5 | 11.5 | 3.5 | 11.4 |
| Qwen2.5-7B-LaMSeI | 46.0 | 50.3 | 12.1 | 21.8 |
| Qwen2.5-7B-LaMSeI-CoT | 59.0 | 63.9 | 15.5 | 25.8 |
| Qwen2.5-14B-DG | 2.0 | 7.0 | 11.0 | 2.7 |
| Qwen2.5-14B-CoT | 25.5 | 30.4 | 16.5 | 19.3 |
| Qwen2.5-14B-LaMSeI | 51.0 | 55.4 | 18.5 | 17.8 |
| Qwen2.5-14B-LaMSeI-CoT | 51.5 | 55.7 | 16.5 | 17.0 |

## D.2    Ablation Study on Model Temperature

The temperature parameter plays a crucial role in controlling the randomness and diversity of the generated responses from the LLM. We have added an ablation experiment to study the influence of temperature, using the model of Qwen-2.5-7B-Instruct. As the results shown in the Tab. 11, a higher temperature would lead to more diverse and random responses, potentially increasing the uncertainty estimation. However, performance changes become less pronounced when the temperature exceeds a threshold (e.g., $> 0.3$).

Table 11: Ablation study on model temperature. The experiments are conducted with Qwen2.5-7B-Instruct.

| Temperature | 2WikiMultiHopQA | MuSiQue |
|---|---|---|
| 0.1 | 31.7 | 20.7 |
| 0.3 | 49.0 | 21.1 |
| 0.5 | 50.3 | 21.8 |
| 0.7 | 57.0 | 20.0 |
| 0.9 | 57.0 | 21.0 |

## D.3    Results on AmbigQA Dataset

We further conduct experiments with AmbigQA dataset Min et al. (2020), which aligns closely with our focus on clarifying user intent through selective interaction, as the results shown in the Tab. 12. The results demonstrate the effectiveness of LaMSeI method on AmbigQA dataset.

Table 12: Experimental results on AmbigQA dataset.

| Method | AmbigQA |
|--------|---------|
| DG | 46.0 |
| CoT | 42.5 |
| LaMSeI | 47.5 |
| LaMSeI+CoT | 51.0 |

### D.4 Combine LaMSeI with Perplexity Measurement

LaMSeI implements uncertainty estimation module via multiple answers sampling, which shares similarities with perplexity measurement. In our study, we opted for multiple answer sampling due to its ability to directly reflect the semantic divergence of model outputs, which aligns well with our research focus on semantic-level uncertainty in language models. Despite that, perplexity-based methods could also be valuable and worthy of further exploration. Thus, we conduct experiment with perplexity-based methods, as shown in Tab. 13. The results show that perplexity measurement can also be incorporated to LaMSeI method, and improves the final answer of LLM compared to DG method.

Table 13: Experiments of LaMSeI with perplexity measurement for uncertainty estimation.

| ES | | | IS | | | TS | | |
|------|------|------|------|------|------|------|------|------|
| Win | Lose | Tie | Win | Lose | Tie | Win | Lose | Tie |
| 52.18 | 34.97 | 12.85 | 45.92 | 40.96 | 13.12 | 44.72 | 37.70 | 17.58 |

### D.5 Combine LaMSeI with New Uncertainty Estimation Method

To further demonstrate the effectiveness of LaMSeI framework, inspired by (Li et al., 2023a), we introduce Instruction-Following Difficulty (IFD) score $\text{IFD}_\theta(Q, A)$ to assess model's confidence, formulated as: $\text{IFD}_\theta(Q, A) = \frac{s_\theta(A|Q)}{s_\theta(A)}$, where $s_\theta(A|Q)$ is the next-token prediction loss of the answer given the question, and $s_\theta(A)$ is the next-token prediction loss of the answer itself. A higher IFD score demonstrates that the model struggles to generate a confident answer given the instruction and question, suggesting the necessity of actively seeking user's input. While a lower IFD score indicates that the model is familiar with the given question, there's no need to inquiry for additional information. We utilize $\tau$ as the IFD score threshold to determine whether to interact with the user and perform ablation experiment on $\tau$ to examine its influence. As shown in Tab. 14, the results align with using $\delta$ as threshold, which means that acquiring more information from the user is beneficial for obtaining the final response.

| $\tau$ | ES | | | IS | | | TS | | |
|--------|------|------|------|------|------|------|------|------|------|
| | **Win** | **Lose** | **Tie** | **Win** | **Lose** | **Tie** | **Win** | **Lose** | **Tie** |
| 0.2 | **48.87** | 14.62 | 36.51 | **45.36** | 15.34 | 30.02 | **43.26** | 18.48 | 38.26 |
| 0.4 | **25.13** | 9.82 | 65.05 | **19.87** | 7.51 | 12.36 | **27.97** | 7.75 | 64.28 |
| 0.6 | **10.29** | 6.36 | 83.35 | **8.90** | 5.67 | 3.23 | **10.68** | 6.37 | 82.95 |

Table 14: Ablation study on the effect of IFD score threshold on the QMSum dataset. We use GPT-4 with evaluation prompt (Fig. 6) to assess the response quality of LaMSeI against DG.

### D.6 More Algorithm Running Examples

We provide more examples of LLM output during LaMSeI running process in Fig. 8, Fig. 9, Fig. 10 and Fig. 11. It is evident that LaMSeI poses useful clarifying questions regarding the ambiguous user query, thereby enhancing the interpretation of user intent.

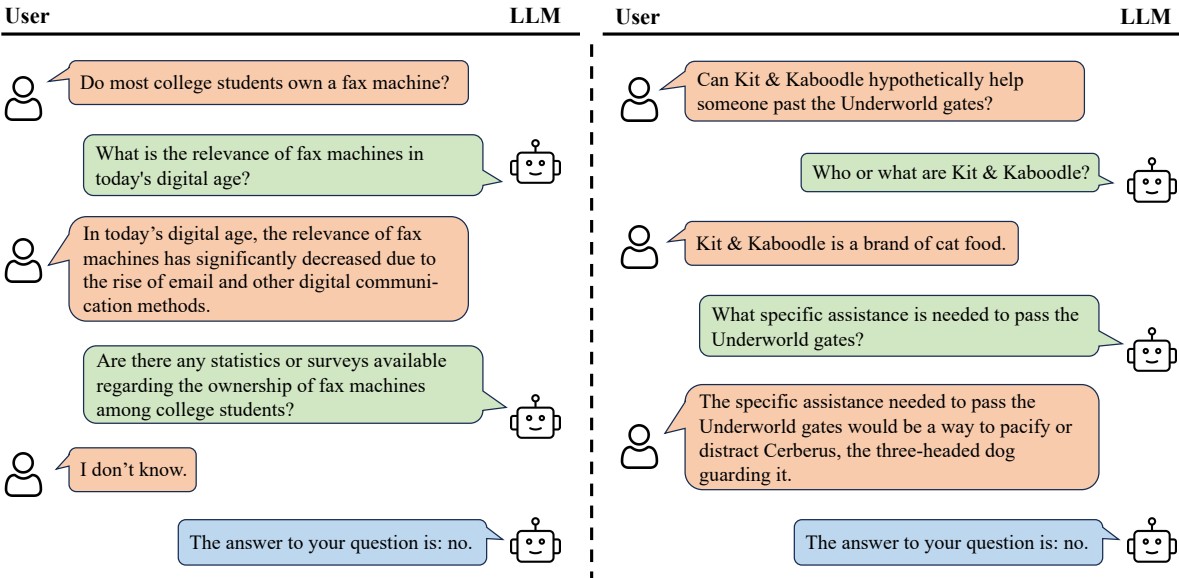

Figure 8: Examples of model output for LaMSeI. The green chat frame presents the clarifying questions that LLM actively inquires the user.

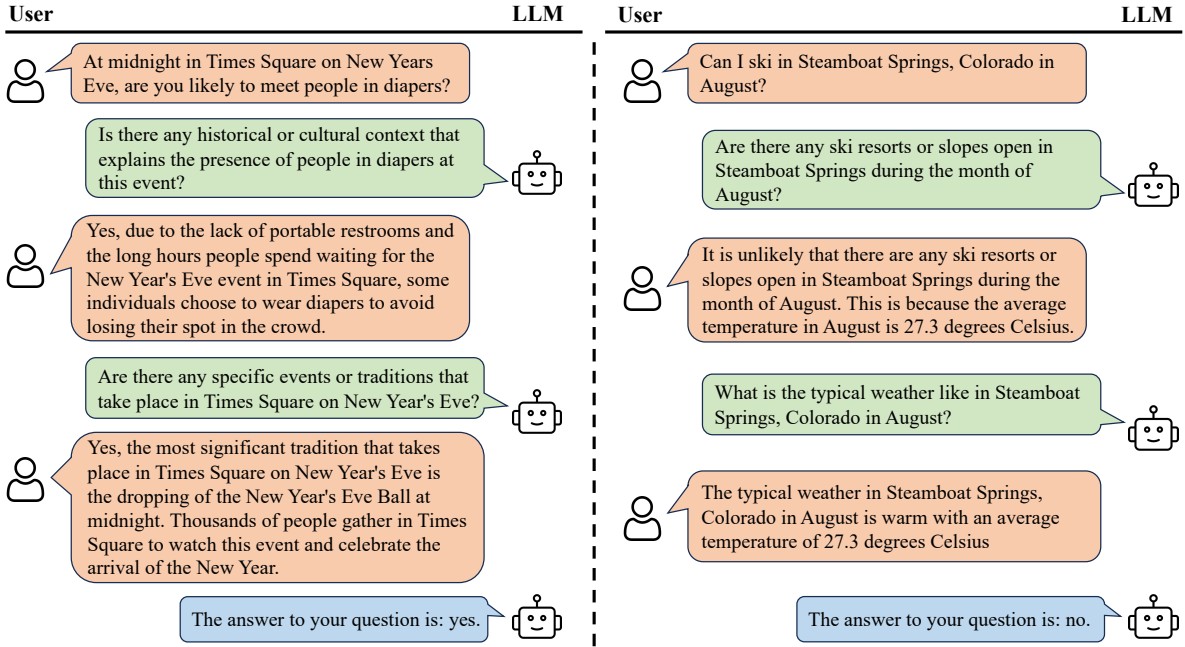

Figure 9: Examples of model output for LaMSeI. The green chat frame presents the clarifying questions that LLM actively inquires the user.

# E  Details about Experiments with Human Involvement

We use the human evaluation as the gold standard to judge whether LaMSeI is effective. We conduct human-participated experiments where actual human feedback is integrated. The experiment is based on the QMSum dataset, from which 100 user queries are randomly selected for evaluation. We invite five volunteers to the experiment, each accounting for 20 queries. Participants respond to clarifying questions posed by the LLM, which subsequently generates answers informed by the human response. Subsequently, another

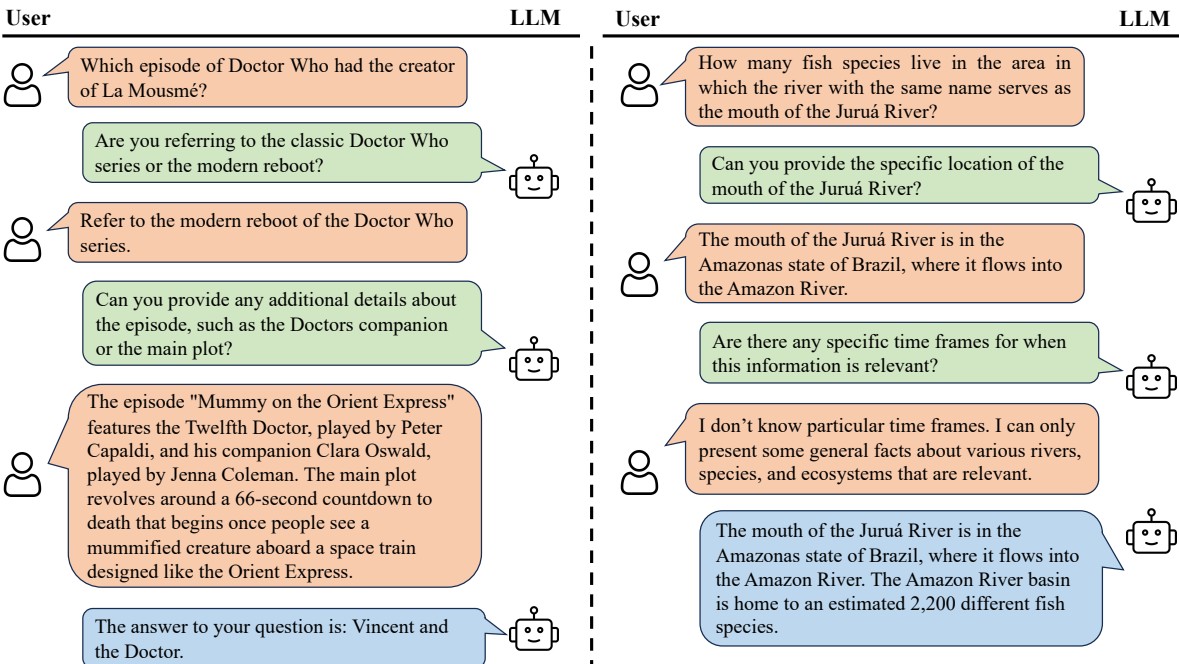

Figure 10: Examples of model output for LaMSeI. The green chat frame presents the clarifying questions that LLM actively inquires the user.

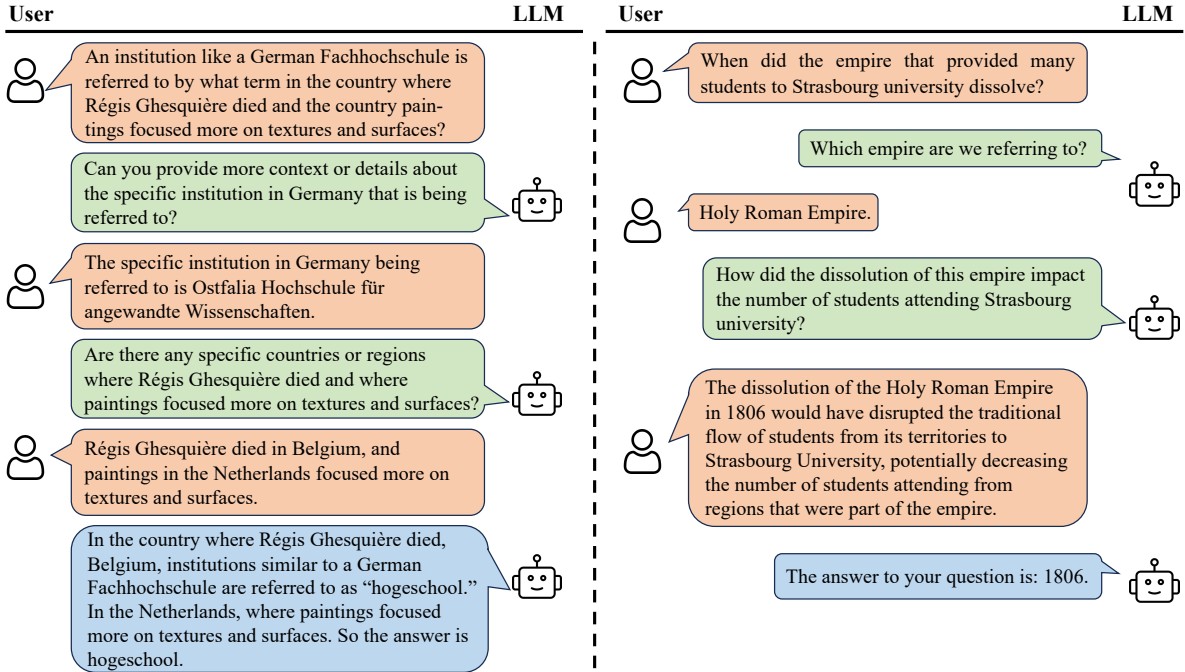

Figure 11: Examples of model output for LaMSeI. The green chat frame presents the clarifying questions that LLM actively inquires the user.

participant compares the answers generated by LaMSeI and DG and annotates a preference for one over the other. During evaluation, each participant is asked with the following annotation: "Which option is better(0 represents the first is better and 1 represents the second and 2 represents a tie):" To avoid the impact of choice sequence, we random the generated answers from LaMSeI and DG. The experimental results

are illustrated in Fig.3(b). The human evaluation results reveal that LaMSeI surpasses DG in 48% of the instances, demonstrating its superior ability to comprehend user input during human interaction. Conversely, DG's responses are preferred in only 23% of the cases, underscoring the effectiveness of LaMSeI in the actual application scenario.

## F    Broader Impact Statement

LaMSeI enhances the reliability of AI systems in high-stakes scenarios like healthcare and education by enabling LLMs to ask clarifying questions only when uncertain, minimizing user burden while improving response accuracy. This capability could reduce risks of misinformation in sensitive contexts (e.g., clinical advice) and foster trust in AI-assisted decision-making. However, errors in uncertainty estimation might lead to overconfident but incorrect responses, necessitating rigorous bias mitigation and transparency in model design. Privacy concerns also arise if clarifying queries inadvertently pressure users to disclose sensitive data, requiring robust anonymization protocols.

While LaMSeI's compatibility with diverse LLMs (e.g., GPT-3.5, LLAMA) broadens its applicability, its computational demands may limit access for resource-constrained communities, exacerbating AI inequity. Empirical results—19% accuracy gains and human-preferred outcomes in 82% of cases—underscore its potential to redefine human-AI collaboration. Future efforts should prioritize ethical safeguards, equitable deployment, and multilingual adaptation to maximize societal benefit and minimize harm.

