# OpenReview forum: "Interactive Large Language Models for Reliable Answering under Incomplete Context"
_TMLR — Accepted by TMLR_

### Review · Reviewer_fgPM · 2025-05-01

**Summary Of Contributions:**

In the question-answering process between humans and LLMs, human queries may sometimes lack necessary information. In such cases, the LLM needs to actively engage in interaction to request the missing details. This work estimates the uncertainty of LLMs by employing Multiple Answers Sampling, thereby categorizing the  responses of LLMs into high uncertainty and low uncertainty. For responses with high uncertainty, the LLM actively initiates interaction by asking selective clarifying questions to obtain more background information, enabling it to provide better answers. Extensive experiments demonstrate the effectiveness of this approach.

**Audience:**

Yes

**Claims And Evidence:**

Yes

**Requested Changes:**

1. The uncertainty estimation via multiple answers sampling essentially resembles a form of perplexity measurement. The authors might consider exploring other perplexity-based methods. For example, it is possible to directly compute the perplexity as follows:
$$
\text{PPL} = \exp\left( - \frac{1}{N} \sum_{i=1}^{N} \log p(w_i \mid w_1, \ldots, w_{i-1}) \right)
$$
where $N$ is the total number of tokens in the evaluation set, and $p(w_i \mid w_1, \ldots, w_{i-1})$ is the predicted probability of the $i$-th token given the preceding context.

2. The authors could consider using a stronger CoT baseline in your experiments. Currently, the implemented CoT uses only the basic zero-shot prompt (“Let’s think step by step”). Incorporating more advanced CoT prompting methods would provide a fairer and more competitive comparison.

**Strengths And Weaknesses:**

Strengths:

1. This work helps enhance the ability of LLMs to determine when interaction is necessary in ambiguous or incomplete contexts, and enables it to autonomously select the most informative questions to effectively uncover missing context.

2. Moreover, this work takes into account the potential burden of interactive questioning on humans. Compared to the greedy asking (GA) method, it achieves better performance with only three clarifying questions, surpassing GA even when it asks five.

Weaknesses:

1. The uncertainty estimation method proposed in this work may not necessarily be the most suitable for this domain, and further experimental validation is needed to confirm its effectiveness.

2. In the baseline, the CoT uses the simplest zero-shot prompting method. Given the wide variety of recent CoT-related approaches, comparing only against the “let’s think step by step” baseline may raise concerns about the fairness of the comparison.

---

> ### Author Response · Authors · 2025-06-05
> **Author Response**
>
> We appreciate your insightful comments. To address your concern, we have included additional experiments. Please find our response below.
>
> > Comment 1: The uncertainty estimation via multiple answers sampling essentially resembles a form of perplexity measurement. The authors might consider exploring other perplexity-based methods.
>
> Response: We acknowledge that the uncertainty estimation method via multiple answers sampling in our paper shares similarities with perplexity measurement. In our study, we opted for this approach due to its ability to directly reflect the semantic divergence of model outputs, which aligns well with our research focus on semantic-level uncertainty in language models. Despite that, we agree that other perplexity-based methods could also be valuable and worthy of further exploration. Thus, we conduct experiment with perplexity-based methods, as shown in below table. The results show that perplexity measurement can also be incorporated to LaMSeI method, and improves the final answer of LLM. We have updated the additional results in Appendix D.4 in the updated version.
>
> |       | ES    |       |       | IS    |       |       | TS    |       |
> | ----- | ----- | ----- | ----- | ----- | ----- | ----- | ----- | ----- |
> | Win   | Tie   | Lose  | Win   | Tie   | Lose  | Win   | Tie   | Lose  |
> | 52.18 | 34.97 | 12.85 | 45.92 | 40.96 | 13.12 | 44.72 | 37.70 | 17.58 |
>
> > Comment 2: The authors could consider using a stronger CoT baseline in your experiments. Currently, the implemented CoT uses only the basic zero-shot prompt (“Let’s think step by step”). Incorporating more advanced CoT prompting methods would provide a fairer and more competitive comparison.
>
> Response:  We clarify that current implementation of CoT has already used demonstration to generate the answer (as shown in the prompt for CoT in Fig. 7). This setup aligns with the practices of many advanced CoT prompting methods and provides a fair and competitive comparison for our proposed LaMSeI method. *We have updated the introduction to CoT method in Sec. 4.1 to make the presentation clearer*.
>
> ------
>
> Thanks again for your time and effort in providing feedback. We are happy to discuss if you had any additional concerns.

---

### Review · Reviewer_Zf6e · 2025-05-04

**Summary Of Contributions:**

The paper introduces LaMSeI (Language Model with Selective Interaction), a novel method designed to improve the reliability of Large Language Models (LLMs) by selectively interacting with users to gather essential missing context. The paper proposes a method to measure the uncertainty of LLM responses via sampling multiple answers and computing semantic variation. Also, the paper uses active learning techniques (similarity- and diversity-based strategies) to select the most informative clarifying questions. The performance on multiple datasets demonstrates the work's effectiveness.

**Audience:**

Yes

**Claims And Evidence:**

Yes

**Requested Changes:**

1. Please clarify the advantage of using a third-party embedding model instead of LLM embeddings.
2. Please clarify why low variance is enough for selecting the correct answers with confidence.
3. Please describe how much the temperature influences the results.
4. Please provide prompts for GPT-4 to simulate humans. Will the personnel in IQA-EVAL help in the interactive question answering process?
5. Please test your performance on the AmbigQA dataset. It is most related to the clarifying.

**Strengths And Weaknesses:**

Strengths:
1. The paper effectively integrates active learning with uncertainty estimation for selective interactivity.
2. It provides a practical solution for necessary clarifications.
3. It demonstrates advantages across various datasets and LLM models.
4. It includes both simulated and human-involved evaluations, enhancing the method's robustness.

Weaknesses:
1. The method relies heavily on third-party embedding models for uncertainty estimation. Why don't you use the embeddings generated by LLMs? You apply the LLAMA series models and should be able to get embeddings directly.
2. Low variance may not indicate correct answers. This method does not have a mechanism to mitigate this type of issue.
3. The answer sample process requires appropriate temperature parameters. No ablation study discusses the effectiveness of different temperatures.
4. No descriptions about the prompt for GPT-4 to simulate humans? Do you set a personal goal for it, as in IQA-EVAL?
5. What is the performance on ambiguousQA? It seems like your method should work on the dataset by multi-turn conversation and the clarifying procedure.

---

> ### Author Response · Authors · 2025-06-05
> **Author Response (Part 1/2)**
>
> We appreciate your highlight of the practicability and effectiveness of the method. Please find our response to each comment as follow.
>
> > Comment 1: Please clarify the advantage of using a third-party embedding model instead of LLM embeddings.
>
> Response: The core reason is that third-party embedding models like text-embedding-ada-002 are often optimized for general-purpose semantic similarity tasks and have been extensively tested for their ability to capture nuanced relationships between texts, which is critical for accurately estimating the uncertainty of our LLM's responses. These specialized embedding models can provide more consistent and comparable vector representations across different queries and answers, which enhances the reliability of our uncertainty estimation metric. Additionally, using a dedicated embedding model allows for computational efficiency, as it can be more lightweight and faster for the specific task of generating embeddings for uncertainty calculation, compared to invoking the full LLM for this purpose. This separation of concerns also enables more straightforward experimentation and comparison across different LLM backbones, as the embedding model remains consistent while the LLM may vary. *We have updated Sec. 3.3 to better clarify the advantage of using a third-party embedding model* in the updated version.
>
> > Comment 2: Please clarify why low variance is enough for selecting the correct answers with confidence.
>
> Response: The LaMSeI method estimates the uncertainty of the LLM regarding the query by sampling multiple responses and calculating their semantic variation. In the paper, low variance indicates that the LLM's multiple responses are semantically consistent, reflecting its high confidence in the query. When uncertainty is low, the LLM can directly generate an answer without further interaction. This approach not only reduces the user's interaction burden but also demonstrates the model's confidence in the query. However, low variance does not guarantee absolute correctness, as uncertainty may still exist in certain cases. The paper uses the low variance threshold as a heuristic to guide the LLM to interact with users only when necessary, thereby improving the efficiency and reliability of the model's responses. *We have updated Sec. 3.2.1 to incorporate the discussions about the model uncertainty*.
>
> > Comment 3: Please describe how much the temperature influences the results.
>
> Response: The temperature parameter plays a crucial role in controlling the randomness and diversity of the generated responses from the LLM. We have added an ablation experiment to study the influence of temperature, using the model of Qwen-2.5-7B-Instruct. As the results shown in the below table, a higher temperature would lead to more diverse and random responses, potentially increasing the uncertainty estimation. However, performance changes become less pronounced when the temperature exceeds a threshold (e.g., > 0.3). We have incorporated the additional results in Appendix D.2.
>
> | Temperature | 2WikiMultiHopQA | MuSiQue |
> | ----------- | --------------- | ------- |
> | 0.1         | 31.7            | 20.7    |
> | 0.3         | 49.0            | 21.1    |
> | 0.5         | 50.3            | 21.8    |
> | 0.7         | 57.0            | 20.0    |
> | 0.9         | 57.0            | 21.0    |
>
> > Comment 4: Please provide prompts for GPT-4 to simulate humans. Will the personnel in IQA-EVAL help in the interactive question answering process?
>
> Response: We appreciate the reviewer's inquiry regarding the prompts and human involvement in our experiments. The prompts used to simulate human interaction with GPT-4 are presented in Fig. 5. The prompts are designed to ensure that GPT-4 responds to clarifying questions based mainly on the supporting facts, without introducing subjective biases or additional context. As for the involvement of IQA-EVAL personnel, our experimental framework does not directly engage IQA-EVAL staff in the interactive question-answering process. Instead, we adopt a simulation approach where GPT-4, acting as a pseudo-human interlocutor, interacts with the LLM based on predefined prompts and datasets. This design allows us to evaluate the effectiveness of LaMSeI in a controlled environment while maintaining experimental reproducibility. Although IQA-EVAL emphasizes human participation, our method focuses on leveraging GPT-4 to simulate human behavior, ensuring that the interaction process remains consistent with the principles of IQA-EVAL while adapting to the specific requirements of our research.

---

> ### Author Response · Authors · 2025-06-05
> **Author Response (Part 2/2)**
>
> > Comment 5: Please test your performance on the AmbigQA dataset. It is most related to the clarifying.
>
> Response: We sincerely appreciate the reviewer’s suggestion to evaluate our method on the AmbigQA dataset, which aligns closely with our focus on clarifying user intent through selective interaction, as the results shown in the below table. The results demonstrate the effectiveness of LaMSeI method on AmbigQA dataset. We have update the Appendix D.3 to incorporate the new results.
>
> | Method     | AmbigQA |
> | ---------- | ------- |
> | DG         | 46.0    |
> | CoT        | 42.5    |
> | LaMSeI     | 47.5    |
> | LaMSeI+CoT | 51.0    |
>
> We believe the experiment and the result analysis have been clearly enhanced based on your comments. If you had any further concerns, please let us know.

---

### Review · Reviewer_tzck · 2025-05-20

**Summary Of Contributions:**

This paper introduces LaMSeI, a method designed to enhance the reliability of Large Language Models (LLMs) when answering questions under incomplete context. Instead of always attempting to direct answer input question, LamSeI first estimate the LLM's uncertainty about the user query, and introduce additional interaction with user when the uncertainty is high. They employ active learning strategies to select the most informative additional question from a LLM-generated candidate question pool. Experimental results show that the proposed method outperforms baselines on a variety of tasks.

**Audience:**

Yes

**Broader Impact Concerns:**

Not much.

**Claims And Evidence:**

Yes

**Requested Changes:**

- Include additional experiments with more advanced LLMs (Qwen-2.5, LLaMA-3.2). It would be more convincing to include model of different sizes to validate the scalability of the method.
- Clarify the setup and interpretation of Section 4.6, particularly the definition and implications of the mask rate.
- Provide a more in-depth analysis of the quality of clarifying questions and their impact on final answer quality.

**Strengths And Weaknesses:**

### Strengths

- The paper is overall clear, well-written and easy to follow.
- The selective interaction framework combining uncertainty estimation and active learning is well-motivated, and is a practical enhancement of LLM deployment.
- Experiments conducted across multiple datasets and base models validate the effectiveness of the proposed method. Ablation studies further support the contribution of individual components.

### Weaknesses

- The base models used in the experiments are somewhat outdated. The paper would benefit from including results on more recent LLMs such as Qwen2.5 or LLaMA-3.2.

- Sec 4.6 is somewhat confusing. As the mask rate increases, the LLM receives less contextual information, yet performance improves. What does mask rate refer to? Why do the win rate increase with higher mask rate?
- The paper lacks a detailed analysis of the quality of LLM-generated clarifying questions, and how the additional user-provided information directly contributes to performance improvements. Do uncertainty in LLM correlate with error rate in generation?

---

> ### Author Response · Authors · 2025-06-05
> **Author Response**
>
> We sincerely appreciate your insightful feedback and finding this work well-motivated. We have added additional experiments and discussions following your suggestions. Below, we answer your questions point-by-point.
>
> > Comment 1: Include additional experiments with more advanced LLMs.
>
> **Response**: We agree and have added results for Qwen-2.5-7B and Qwen-2.5-14B on 2WikiMultiHopQA and MuSiQue datasets, as shown in the below table. Overall, LaMSel can also improve these advanced models’ performance, outperforming baselines such as DG and CoT, confirming LaMSeI’s effectiveness on more recent models. *We have updated Appendix D.1 to incorporate the additional results in the updated version*.
>
> | Model                | Method     | 2WikiMultiHopQA | MuSiQue |
> | -------------------- | ---------- | --------------- | ------- |
> | Owen2.5-7B-Instruct  | DG         | 21.5            | 9.6     |
> |                      | CoT        | 11.5            | 11.4    |
> |                      | LaMSeI     | 50.3            | 21.8    |
> |                      | LaMSeI+CoT | 63.9            | 25.8    |
> | Owen2.5-14B-Instruct | DG         | 7.0             | 2.7     |
> |                      | CoT        | 30.4            | 19.3    |
> |                      | LaMSeI     | 55.4            | 17.8    |
> |                      | LaMSeI+CoT | 55.7            | 17.0    |
>
> > Comment 2: Clarify the setup and interpretation of Section 4.6, particularly the definition and implications of the mask rate.
>
> **Response**: We appologize for unclear representation. We clarify that mask rate denotes the *percentage of supporting facts removed* from the input context. We implement the context mask by directly masking the tokens of support facts that are exposed to the LLM. Higher mask rates increase ambiguity, allowing LaMSel to demonstrate stronger relative gains by actively seeking missing information. Win rates rise because LaMSel’s interaction compensates for information loss, while baselines fail. *We have updated Sec. 4.6 to clarify the meaning of mask rate*.
>
> > Comment 3: Provide a more in-depth analysis of the quality of clarifying questions and their impact on final answer quality.
>
> **Response**: Current manuscript has presented a preliminary analysis on the quality of clarifying questions. To be more specific, the quality of clarifying questions directly determines whether LaMSel successfully resolves ambiguity and generates accurate answers. In the *success case*, the selected questions (e.g., *"Are there specific movements in waltz/slam dancing that could lead to injuries?"*) precisely target the core of the user query ("*Is waltz less injurious?*"). These questions align with the supporting facts (e.g., slam dance involves "collisions"), enabling LaMSel to extract critical context about injury mechanisms and produce a correct answer. Conversely, in the *failure case*, the selected questions (e.g., *"Other income sources for Billie Eilish?"* or *"Average Porsche price?"*) are misaligned with the query's key uncertainty ("*Can she afford it?*"). They overlook the decisive evidence in the supporting facts, leading to an incorrect answer despite sufficient information being present. This contrast confirms that high-quality questions must directly address the information gap implied by the query and supporting context; irrelevant or overly broad questions fail to reduce uncertainty and degrade answer accuracy. Future work could better leverage this feature to generate clarify questions. *We have update Sec. 4.3 to incorporate the in-depth discussions*.
>
> ------
> We hope that our response has addressed your concerns satisfactorily. If you had any further concerns, we are glad for discussion.

---

### Author Response · Authors · 2025-06-05
**General Response**

Dear Editor and Reviewers,

Thank you for your valuable feedback on our manuscript. We appreciate the reviewers' recognition of the work's motivation (Reviewer tzck), practicability (Reviewer Zf6e), and the experimental sufficiency/performance (Reviewers tzck, Zf6e, fgPM). We have carefully addressed all comments through new experiments, clarifications, and expanded discussions. The [revised manuscript](https://openreview.net/pdf?id=nnlmcxYWlV) incorporates these changes, with updates highlighted in red. Below is a summary of revisions:
1. Enhanced Experiments (Reviewer tzck,Zf6e,fgPM):
- Add results for Qwen-2.5-7B/14B (Appendix D.1).
- Conduct experiments on AmbigQA dataset (Appendix D.3).
- Perform temperature ablation and analysis (Appendix D.2) and perplexity-based uncertainty comparisons (Appendix D.4).
2. Clarifications (Reviewer tzck,Zf6e,fgPM):
- Explicitly define the "mask rate" (Sec. 4.6).
- Explain that third-party models enable efficient, reliable uncertainty estimation (Sec. 3.3).
- Clarify that the CoT baseline uses step-by-step reasoning, aligning with advanced prompting (Sec. 4.1).
3. Expanded Analysis (Reviewer tzck,Zf6e):
- Deepen analysis of clarifying questions' impact on answer quality (Sec. 4.3).
- Further explain uncertainty estimation for interaction (Sec. 3.2.1).

------
Thank you again for your time and insights—we welcome further feedback.

---

### Comment · Editors_In_Chief · 2025-08-27

The EiCs updated the camera-ready version of this paper on August 26, 2025, at the request of the authors. This update corrects the affiliation of one of the authors.

---

### Decision · Action_Editor_mEJz · 2025-06-20

**Recommendation:** Accept with minor revision

**Audience:**

Yes

**Audience Explanation:**

The paper deals with Large Language Models, reliability, uncertainty estimation, active learning, and question answering – all of which are central topics in machine learning and natural language processing research, which TMLR covers. All three reviewers (fgPM, Zf6e, tzck) explicitly answered "Yes" to the question "Audience: Yes" in their official recommendations, indicating they believe the paper is suitable for the TMLR audience.

**Claims And Evidence:**

Yes

**Claims Explanation:**

The initial consensus was that the claims were supported. The reviewers then pinpointed areas where the evidence could be improved in terms of accuracy (e.g., hyperparameter effects), convincingness (e.g., newer models, additional datasets, stronger baselines), and clarity (e.g., definitions, methodological rationale). The authors' revisions systematically addressed these points by adding new experiments, clarifications, and expanded discussions. Therefore, after the revision process, the claims are backed by more robust, comprehensive, and clearly presented evidence.